# Evaluating LLMs for Combinatorial Optimization: One-Phase and Two-Phase Heuristics for 2D Bin-Packing

## Abstract

This paper presents an evaluation framework for assessing Large Language Models' (LLMs) capabilities in combinatorial optimization, specifically addressing the 2D bin-packing problem. We introduce a systematic methodology that combines LLMs with evolutionary algorithms to generate and refine heuristic solutions iteratively. Through comprehensive experiments comparing LLM generated heuristics against traditional approaches (Finite First-Fit and Hybrid First-Fit), we demonstrate that LLMs can produce more efficient solutions while requiring fewer computational resources. Our evaluation reveals that GPT-4o achieves optimal solutions within two iterations, reducing average bin usage from 16 to 15 bins while improving space utilization from 0.76-0.78 to 0.83. This work contributes to understanding LLM evaluation in specialized domains and establishes benchmarks for assessing LLM performance in combinatorial optimization tasks.

## 1 Introduction

The evaluation of Large Language Models (LLMs) extends beyond traditional natural language processing tasks to specialized domains like combinatorial optimization. The 2D bin-packing problem that is placing rectangles into the minimum number of fixed-size bins represents a challenging NP-hard optimization task that serves as an ideal testbed for evaluating LLM capabilities in mathematical reasoning and algorithmic design.

Traditional heuristic approaches like Finite First-Fit (FFF) and Hybrid First-Fit (HFF) provide established baselines, but their performance limitations in scalability and solution quality create opportunities for LLM enhanced approaches. This paper evaluates how effectively LLMs can generate, refine, and optimize heuristic algorithms through an iterative evolutionary framework.

Our evaluation framework addresses key questions: Can LLMs understand complex algorithmic constraints? How do LLM generated solutions compare to established heuristics? What evaluation metrics best capture LLM performance in optimization contexts?

## 2 Mathematical Formulation

The two-dimensional bin packing problem (2D-BPP), an NP-hard problem, seeks to pack $n$ items of size $(w_i, h_i)$ into the minimum number of bins of size $(W, H)$, where $W > w_i$ and $H > h_i$ for all $i \in \{1, \ldots, n\}$ Wäscher et al. (2007); Martello & Toth (1990); Garey & Johnson (1979).

Let the indicator variable $z_{ij} = 1$ when item $i$ is placed in bin $j$ and 0 otherwise; similarly, $u_j = 1$ when bin $j$ is used and 0 otherwise.

By the pigeonhole principle, a maximum of $n$ bins is needed Johnsonbaugh (2018). The optimization problem is formulated as follows:

$$\min \sum_{j=1}^{n} u_j$$

Subject to, for all $i, j \in \{1, \ldots, n\}$:

$$\sum_{j=1}^{n} z_{ij} = 1,$$
$$0 \leq x_{ij} \leq (W - w_i)z_{ij},$$
$$0 \leq y_{ij} \leq (H - h_i)z_{ij},$$
$$u_j \geq z_{ij},$$
$$\text{(non-overlap constraints).}$$

Finally, the total utilization, a common metric to evaluate performance for a given solution, is measured as

$$\rho_{\text{total}} = \frac{\sum_{i \in I} w_i h_i}{\left(\sum_{j=1}^{n} u_j\right) WH} Ioriet\ al.\ (2021);\ Oliveiraet\ al.\ (2023).$$

## 3 METHODOLOGY

**Problem Formulation and Constraints:** We evaluate LLMs on the 2D bin-packing problem with strict constraints: bin dimensions of 200×100 units, item constraints requiring no overlap and complete containment within bins, the objective to minimize number of bins used, and an evaluation dataset of 50 randomly generated squares (10-50 units) across 20 iterations.

**LLM Based Evolutionary Process:**

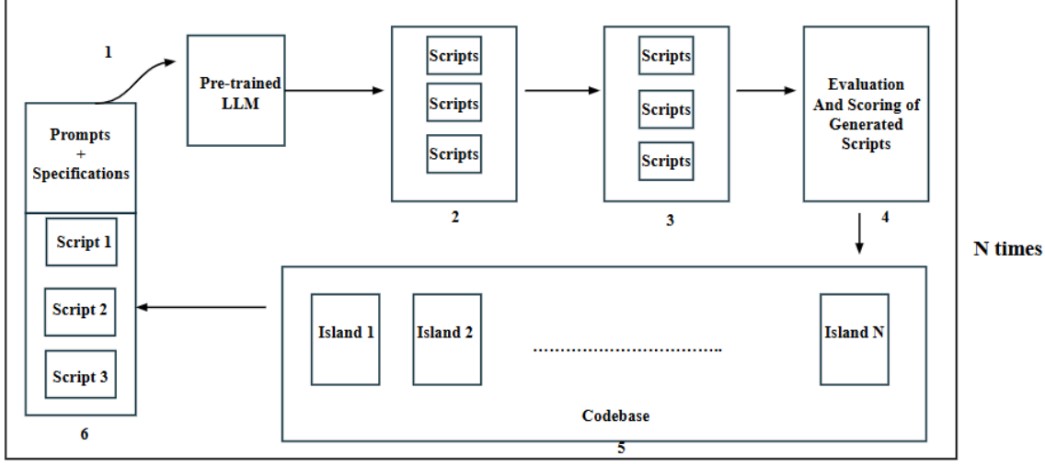

Figure 1: Iterative Evolutionary Framework for Heuristic Generation. Please zoom in for detail.

Our evaluation methodology employs a six-step iterative process. First, structured prompting designs prompts that clearly specify problem constraints, input/output formats, and success criteria. Second, code generation and correctness validation systematically validates LLM generated candidate solutions against constraint satisfaction. Third, performance scoring evaluates solutions using multiple metrics: number of bins used (primary), space utilization efficiency (secondary), and execution time (tertiary). Fourth, island-based selection clusters high-performing solutions into "islands" to promote diversity. Fifth, iterative refinement uses the top performing solutions to inform subsequent prompts, creating an evolutionary feedback loop.

To implement this framework, each generated script is rigorously validated for syntactic and logical correctness; only solutions that successfully pack all items according to the rules are advanced to the performance evaluation stage. The high performing solutions are clustered into distinct "islands" to preserve strategic diversity and prevent premature convergence on a single type of solution. In the refinement stage, the top three performing solutions one from each of the top three islands are used

as "best-shot" examples in the prompt for the next generation cycle. This evolutionary feedback loop instructs the LLM to learn from the most successful strategies, progressively enhancing the quality of the generated heuristics over six full iterations.

**Baseline Comparisons:** We establish baselines using two established heuristics. Finite First-Fit (FFF) places items in the first available position using First-Fit Decreasing Height (FFDH) with time complexity $O(n^2)$. Hybrid First-Fit (HFF) employs a two-phase approach combining strip packing (FFDH) with bin packing (FFD) with time complexity $O(n \log n)$.

## 3.1 GOAL

The goal of this study is to explore how a Large Language Model (LLM) can autonomously generate, evaluate, and refine heuristics for solving the 2D Bin Packing Problem (2D-BPP). This is accomplished through an iterative loop in which the LLM, specifically GPT-4o, writes Python functions based on a well-defined prompt, evaluates their ability to pack items efficiently, and leverages the best-performing scripts to guide the next round of generation.

Through this multi-round learning framework, we aim to determine whether an LLM can discover a packing strategy that rivals or surpasses classical heuristics like Finite First-Fit (FFF) and Hybrid First-Fit (HFF). This methodology not only examines the end results but also provides insight into how the heuristics evolve through contextual learning and prompt refinement.

## 3.2 DATASET

Each iteration employs a dataset containing 50 randomly generated square items. Each item has a side length randomly chosen between 10 and 50 units. All bins are of fixed dimensions—200 units in width and 100 units in height—and every square must be placed without overlapping and within the confines of the bin. Twenty different datasets were created, each representing a unique and random configuration to simulate varied real-world packing scenarios. By ensuring that each dataset is distinct, the evaluation of heuristic performance remains unbiased and avoids overfitting to any specific pattern of item sizes or arrangements.

## 3.3 FFF AND HFF SCRIPTS

To provide a baseline for performance comparison, two traditional heuristics were implemented: Finite First-Fit (FFF) and Hybrid First-Fit (HFF). In the FFF method, items are sorted by height and packed into the first available bin from bottom to top. If no available position is found within existing bins, a new bin is opened. This greedy strategy is computationally efficient but often results in poor space utilization.

In contrast, the HFF approach operates in two phases. First, it applies the First-Fit Decreasing Height (FFDH) method to create horizontal strip packings by sorting items based on height. Second, these strips are packed into bins using the First-Fit Decreasing (FFD) approach. The combination of strip-level optimization and bin-first fit allocation allows HFF to improve upon the naive nature of FFF, particularly in scenarios involving large numbers of irregular-sized items. Both heuristics were implemented in Python and tested across the same datasets as the LLM-generated heuristics to ensure fair comparison.

### 3.3.1 FINITE FIRST-FIT (FFF) FLOW

---

**Algorithm 1** Finite First-Fit (FFF)

---

1: Initialize empty bins
2: **for** each item **do**
3:     **for** each bin **do**
4:         **if** item fits in bin **then**
5:             Place item in bin
6:             **break** (go to next item)
7:         **end if**
8:     **end for**
9:     **if** no bin fits **then**
10:         Open new bin
11:         Place item in new bin
12:     **end if**
13: **end for**

---

### 3.3.2 HYBRID FIRST-FIT (HFF) FLOW

---

**Algorithm 2** Hybrid First-Fit (HFF)

---

1: Initialize empty bins
2: **for** each item **do**
3:     Select candidate bin using heuristic (First-Fit or alternative)
4:     **if** item fits in selected bin **then**
5:         Place item
6:     **else**
7:         Open new bin
8:         Place item in new bin
9:     **end if**
10: **end for**

---

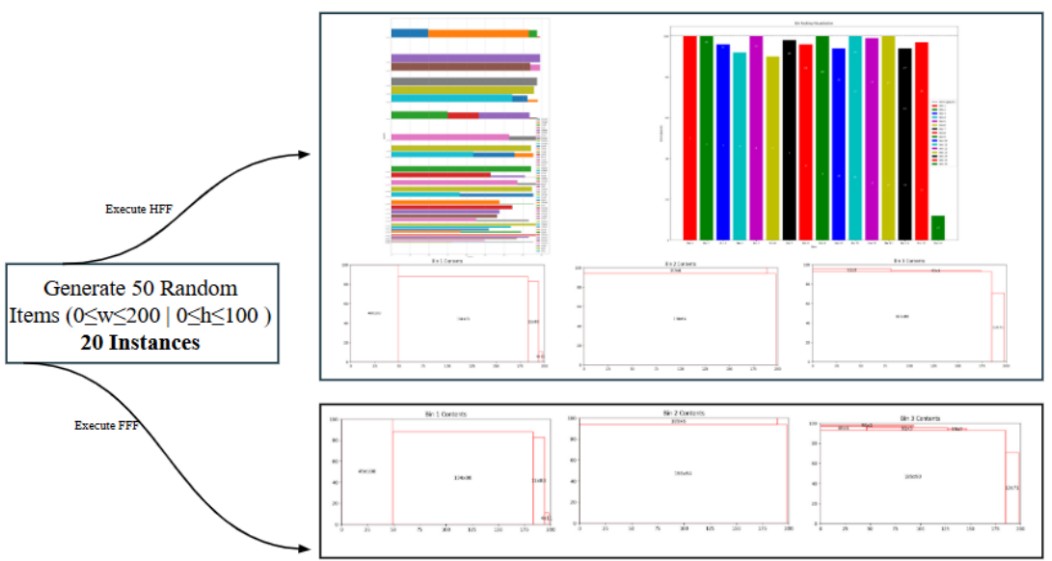

Figure 2: Flowchart of FFF and HFF process steps. Please zoom in for detail.

## 3.4 LARGE LANGUAGE MODEL

We utilized the GPT-4o model from OpenAI for the heuristic script generation. The model was provided with a comprehensive prompt, which included a function prototype, input/output specifications, and strict constraints. The function was expected to accept a NumPy array of item dimensions and a tuple representing bin capacities, and return a list of bins with items mapped to specific coordinates. The prompt explicitly instructed the model to ensure non-overlapping placements, respect bin boundaries, and avoid duplication of items across bins. A template function was included to enforce syntactic consistency, making it easier to validate, test, and compare the generated scripts.

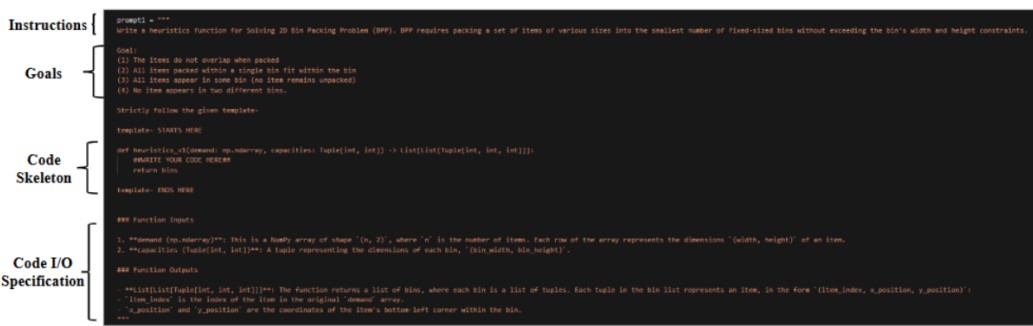

Figure 3: Template used in the LLM prompt. Please zoom in for detail.

## 3.5 PROMPT DESIGN

The prompt is the primary interface used to communicate the 2D Bin Packing Problem to the LLM. It was carefully crafted to ensure the model understood the objective and constraints of the task. The function needed to handle an array of items and place them into bins in such a way that the constraints were strictly satisfied. To guide the LLM effectively, we included a complete Python function signature, detailed descriptions of the expected inputs and outputs, and the rules of the problem. We also clarified the goals, such as minimizing the number of bins and ensuring items did not overlap or exceed bin boundaries. The template helped enforce a consistent structure for all generated heuristics.

The prompt is the main interface between the user and the LLM. We designed a clear and structured prompt to instruct the model to write a heuristic function for solving the 2D Bin Packing Problem (2D-BPP). The prompt defined the packing goals and strictly enforced input-output formats. Each function had to pack a list of rectangular items into bins of fixed size while following strict rules: no item could appear in more than one bin, no items could overlap, and all items had to be packed within the bin boundaries. A template format for the Python function was provided to ensure uniformity across all generated scripts.

## 3.6 SCRIPT GENERATION

The LLM generated multiple scripts based on the initial prompt. Each script was designed to solve the same problem using different logic. A total of 20 scripts were produced during the first round. The variety in the scripts helped cover a wide range of heuristic strategies. These scripts showed significant differences in terms of logic structure, item placement order, and how space within the bins was utilized. Each script was saved for correctness checking and performance scoring.

## 3.7 CORRECTNESS VERIFICATION

Once the scripts were generated, they were tested for correctness. Each script had to meet all the packing constraints. The scripts were run on a fixed set of test cases. Outputs were checked to ensure that no item overlapped, every item was placed within bin boundaries, and no item appeared more than once. Incorrect scripts were discarded. Only those that passed all correctness tests were selected for further evaluation.

## 3.8 SCORING AND EVALUATION

Scripts that passed the correctness tests were scored using the same metrics applied to traditional heuristics. These included the number of bins used, the total packing density, and the script's runtime. Scripts that used fewer bins and maintained higher densities received higher scores. Execution time was also recorded, though it had a lower weight in score calculation. This method ensured that only efficient and practical scripts moved forward.

## 3.9 ISLAND FORMATION

After scoring, high-performing scripts were grouped into islands. Each island contained scripts with similar logic and performance. The term "island" refers to a group of solutions that evolved in parallel but independently. These islands allowed us to preserve diversity among strategies and prevented convergence to a single logic too early in the process.

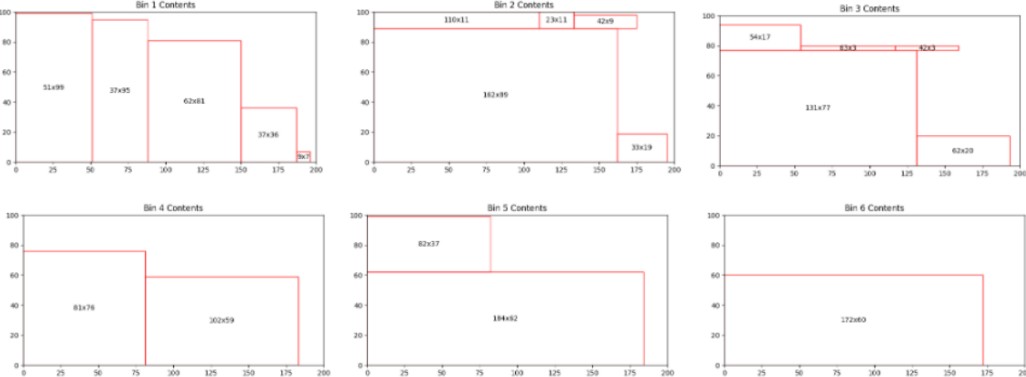

Figure 4: Island distribution after the first iteration, sorted by bin usage. Please zoom in for detail.

## 3.10 ITERATIVE PROMPT REFINEMENT

The top three performing islands were selected to refine the next round of prompts. One script was randomly chosen from each of these top islands. These scripts were embedded into a new prompt as examples. The prompt instructed the LLM to learn from these three solutions and generate a new heuristic function. This process guided the LLM to focus on effective strategies while still producing novel variations. This approach is known as best-shot learning. It helps the LLM improve script quality without losing diversity.

## 3.11 ITERATION AND JUSTIFICATION

The process described above was repeated for six iterations. In each round, the best scripts were selected and used to guide the next generation. The goal was to steadily improve solution quality with each iteration. Six rounds were chosen based on resource availability and diminishing returns. Beyond six rounds, improvements were small compared to the extra cost in time and computation. This number of iterations proved effective in reaching high-performing solutions without excessive overhead. The final scripts from the sixth iteration were used in the evaluation and comparison stages.

## 4 EXPERIMENTAL SETUP:

We conducted experiments using GPT-4o with BPE tokenization on an Intel Core i5-8250U processor with 8GB RAM. The dataset consisted of 20 iterations with 50 randomly generated squares per iteration, and the evaluation protocol used the same dataset for all methods to ensure fair comparison. The LLM evaluation process terminated after demonstrating convergence within 2-6 iterations, indicating rapid solution optimization capability.

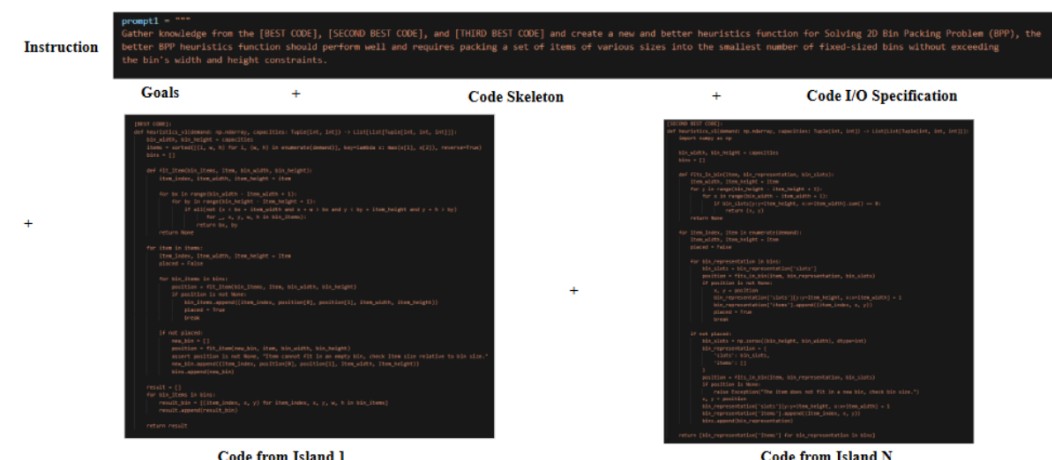

Figure 5: Refined prompt that includes best code samples from top islands. Please zoom in for detail.

## 5 RESULTS AND DISCUSSION

**Comparative Performance**

| Method | Avg Bins | Execution Time (s) | Space Utilization |
|--------|----------|--------------------|--------------------|
| FFF | 16.05 | 0.002446 | 0.76 |
| HFF | 16.00 | 0.024438 | 0.78 |
| LLM | 15.00 | 0.0103 | 0.83 |

Table 1: Comparative performance across evaluation metrics

The LLM-generated heuristic demonstrates superior performance across all evaluation metrics, achieving a 6.25% reduction in bin usage compared to baselines, a 6.4% improvement in space utilization over HFF, and competitive execution time despite code generation overhead.

**Convergence Analysis** The LLM achieved optimal solutions within two iterations, suggesting efficient learning from constraint feedback. This rapid convergence indicates strong pattern recognition capabilities and effective constraint satisfaction learning.

**Space Utilization Patterns** LLM generated solutions show more consistent space utilization across bins (83% average) compared to traditional heuristics, which exhibit declining utilization in later bins (HFF: 86.83% → 63.54%, FFF: 87.50% → 68.00%).

**LLM Capabilities Assessment** Our evaluation reveals several key capabilities. LLMs successfully internalize complex geometric and logical constraints, demonstrating sophisticated constraint understanding. Generated solutions exhibit optimization intuition through sophisticated packing strategies not explicitly programmed. The results show consistent iterative improvement across evolutionary cycles, indicating effective learning mechanisms.

**Limitations and Evaluation Challenges** Computational constraints limit iteration cycles due to API costs, constraining comprehensive evaluation. LLM non-determinism complicates reproducibility, requiring multiple evaluation runs for statistical validity. The evaluation was limited to moderate problem sizes, and larger instances may reveal different performance characteristics that could affect generalization.

**Evaluation Metric Considerations** Traditional optimization metrics (bin count, space utilization) prove effective for LLM evaluation, but additional metrics considering code quality, algorithmic sophistication, and constraint satisfaction robustness could provide deeper insights into LLM problem-solving capabilities.

**Implications for LLM Evaluation** This work contributes to LLM evaluation methodology through domain-specific benchmarking that demonstrates the value of specialized evaluation frameworks for assessing LLM capabilities beyond language tasks. The iterative evaluation protocols show how evolutionary feedback can systematically evaluate LLM learning and adaptation capabilities. Multi-metric assessment establishes that comprehensive LLM evaluation requires performance, efficiency, and solution quality metrics. Finally, baseline establishment provides benchmarks for future LLM evaluation in combinatorial optimization contexts.

## 6 RELATED WORK

The 2D bin packing problem is a fundamental NP-hard combinatorial optimization challenge where rectangular items must be packed into the minimum number of identical bins without overlapping while respecting bin boundaries Wu et al. (2023). Traditional approaches are broadly categorized into one-phase and two-phase algorithms, each offering distinct advantages for different problem scenarios.

One-phase algorithms pack items directly into bins using strategies such as next-fit, first-fit, and best-fit methods combined with placement heuristics like bottom-left (BL) and bottom-left-fill (BLF) to determine specific item positions within selected bins Wu et al. (2023). These approaches prioritize computational efficiency but may sacrifice solution quality due to their greedy nature.

Two-phase algorithms decompose the packing process into sequential stages, with the most established approach using level-based packing where items are first organized into levels of infinite-height strips, then stacked into finite bins Blum et al. (2012). Classic implementations include Hybrid First-Fit (HFF) and Finite Best-Strip (FBS), which build upon foundational algorithms like First-Fit Decreasing Height (FFDH) and Best-Fit Decreasing Height (BFDH) Blum et al. (2012). Modern two-phase approaches have evolved to include sophisticated decomposition strategies such as the Positions and Covering (P&C) methodology, which generates valid item positions before using set-covering formulations for optimal configuration selection Cid-García & Ríos-Solís (2020).

Performance analysis reveals significant trade-offs between solution quality and computational efficiency. Ferreira's comparative study of constructive First-Fit Decreasing strategies, local search, simulated annealing, and genetic algorithms demonstrated that while constructive heuristics provide rapid solutions, improvement-based methods offer superior solution quality at increased computational cost Ferreira (2017). Specific placement strategies like BLF position items iteratively from the lower-left corner, while FFD and BFD algorithms employ different bin selection criteria based on item ordering and space utilization Pintea et al. (2012).

Recent developments have integrated machine learning techniques with traditional heuristics, including deep reinforcement learning approaches for dynamic scenarios and hierarchical frameworks combining heuristic search with learning-based optimization Lee & Nam (2025). However, these approaches remain largely problem specific and have not established systematic evaluation frameworks for assessing algorithmic performance across diverse problem instances. Though the use of LLMs in an evolutionary loop has shown significant promise, for instance, Romera-Paredes et al. Romera-Paredes et al. (2024) introduced FunSearch, a method that pairs an LLM with an evaluator to discover novel, high-performing heuristics for problems such as online bin packing.

Inspired from the work of FunSearch, we contribute to this landscape by introducing a structured evaluation methodology specifically designed for assessing Large Language Model capabilities in generating and optimizing heuristic algorithms for the 2D bin packing problem, addressing the gap in systematic evaluation approaches for AI enhanced combinatorial optimization.

## 7 CONCLUSION

This paper presents a systematic framework for evaluating LLMs in combinatorial optimization contexts. Through comprehensive experiments on the 2D bin-packing problem, we demonstrate that LLMs can generate superior heuristic solutions compared to established algorithms while providing efficient performance. The evaluation framework contributes to understanding LLM capabilities in specialized domains and establishes methodological approaches for assessing LLM performance in optimization tasks.

Our results indicate that LLMs possess significant potential for enhancing combinatorial optimization approaches, achieving measurable improvements in solution quality and computational efficiency. These findings support continued research into LLM applications in mathematical and algorithmic domains while highlighting the importance of rigorous evaluation frameworks for assessing such capabilities.

## 8 FUTURE WORK

Several key research directions emerge from this evaluation framework. First, scalability assessment should investigate how these results scale to larger bin-packing instances or different constraint ratios, as the current 200x100 bins with 10-50 unit squares represents a specific problem space that may not generalize to industrial-scale applications. Second, solution interpretability analysis should characterize the specific strategies the LLM discovered that led to improved performance, as understanding the algorithmic innovations behind the 6.25% improvement would inform future heuristic design and provide insights into LLM reasoning capabilities. Third, reproducibility analysis must address how evaluation frameworks should handle LLM non-determinism through protocols for multiple trial runs, confidence interval reporting, and statistical significance testing to ensure robust evaluation methodologies. Finally, prompt engineering sensitivity requires systematic investigation of how results vary with prompt modifications, as the evaluation framework's dependence on prompt design necessitates establishing reproducibility guidelines and optimal prompting strategies for optimization contexts.

## REPRODUCIBILITY STATEMENT

Every effort has been made to ensure the work presented in this paper is reproducible. A complete source code repository including datasets has been prepared to accompany this work.

## USE OF LLMS

The text presented in this work was presented, in sections, via API to the OpenAI o4-mini model, with the following prompt: "Describe any typos that may be present in the text."

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

## A APPENDIX - SOURCE CODE AND DATASETS

The code and datasets are open source and kept anonymous due to submission policy.

