# OpenReview forum: "Evaluating LLMs for Combinatorial Optimization: One-Phase and Two-Phase Heuristics for 2D Bin-Packing"
_ICLR.cc/2026/Conference — ICLR 2026 Conference Desk Rejected Submission_

### Official Review · Reviewer_HrgV · 2025-10-27

**Soundness:** 2
**Presentation:** 2
**Contribution:** 1
**Rating:** 2
**Confidence:** 4

**Summary:**

This paper presents an evaluation framework to assess Large Language Models (LLMs) in solving the 2D bin-packing problem. The framework combines LLMs with an evolutionary algorithm: LLMs generate heuristic solutions via structured prompting, which are then validated for correctness, scored by metrics like bin usage and space utilization, clustered into "islands" to preserve diversity, and used to refine subsequent prompts iteratively. The LLM’s effectiveness is demonstrated through experiments showing it outperforms traditional heuristics (Finite First-Fit, Hybrid First-Fit)  in the 2D bin-packing problem.

**Strengths:**

1. Propose an new "LLM + evolutionary algorithm" framework for 2D bin-packing problem.
2. Propose a multi-dimensional evaluation system (average bin usage, space utilization, execution time).
3. Use the "island clustering" mechanism to balance strategy diversity and optimization convergence effectively.
4. Good writing and easy to understand.

**Weaknesses:**

1. Lacks novelty in the proposed framework, as the workflow is common used in recent LLM-based automatic algorithm design(e.g. AEL, Funsearch, EoH, ReEvo).
2. Experimental comparisons are limited to two simple heuristics(Finite First-Fit, Hybrid First-Fit), excluding mathematical programming methods, learning-based optimization methods and LLM-based methods.
3. Mentions complex constraints but only tests basic constraints of 2D bin-packing in practical experiments.
4. Incomplete references, only discussed Funsearch for LLM-based methods while omitting relevant works like EoH, ReEvo and so on, and does not mention any mathematical programming methods and learning-based optimization methods.
5. Attached figures (e.g., Figures 3, 4, 5) are unclear.

**Questions:**

1. The proposed framework is kind of Prompt Enginneering and the novelty is poor, what is the key contributions of this work?
2. Only the GPT-4o model is tested, what about other closed-source or open-source large language models(Claude, Gemini, Qwen, DeepSeek)?
3. Mentions that the framework could find the optimal algorithm in just two iterations, whether the framework is secondary to the model's inherent capabilities?
4. Experiments rely on randomly generated data from the same distribution rather than real-world data, leading to insufficient practical significance and potential poor generalization of the generated algorithm.
5. The specific logic of "island clustering" is vaguely described, such as how to define "strategy differences" for island division, how to group new scripts into current island?.

---

### Official Review · Reviewer_iAH4 · 2025-10-27

**Soundness:** 2
**Presentation:** 2
**Contribution:** 2
**Rating:** 2
**Confidence:** 4

**Summary:**

This paper presents a framework that uses an evolutionary process with GPT-4o to generate heuristics for the 2D bin-packing problem. The method involves iterative cycles of code generation, validation, and refinement. The authors demonstrate that their best LLM-generated heuristic outperforms two traditional baselines, finite first-fit (FFF) and hybrid first-fit (HFF).

**Strengths:**

The domain evaluation of LLM-driven heuristic design is an important topic.

**Weaknesses:**

1.The comparison is limited. To better validate the results, please include more diverse and competitive baselines beyond the two hand-crafted ones.
2. The motivation and methodology are not clearly presented.
3. The figures are difficult to read, as the details remain unclear even when magnified.

**Questions:**

Many LLM-driven automated heuristic design methods, including EoH and ReEvo, have been proposed in the last three years. The authors have not compared or discussed any of these baselines. The island-based approach has already been used by FunSearch and many recent works. What is the difference and the advantages of the method used in this paper?

If the authors' focus is not on new methods, the paper still suffers from a lack of details and experimental studies:
i) The methodology for clustering scripts into "islands" based on "similar logic" is vague. What specific features (e.g., code structure, sorting criteria, placement rules) were used to determine similarity?
ii) Only two simple hand-crafted heuristics are compared. You can refer to the settings and baselines used in this paper [3]
iii) The evaluation uses only square items. How would the LLM-generated heuristics perform with more complex, rectangular items, which is a more general and challenging case for 2D-BPP?

[1] Evolution of heuristics: Towards efficient automatic algorithm design using large language model. ICML
[2] Reevo: Large language models as hyper-heuristics with reflective evolution. NeurIPS
[3] Re-evaluating LLM-based Heuristic Search: A Case Study on the 3D Packing Problem. arXiv.

---

### Official Review · Reviewer_Ngnb · 2025-11-01

**Soundness:** 1
**Presentation:** 2
**Contribution:** 1
**Rating:** 0
**Confidence:** 5

**Summary:**

The submission describes an evaluation framework for the ability of a LLM to solve 2d bin packing problem. One-phase and two-phase methods are considered. LLM-generated heuristics are claimed to outperform  human-designed baselines.

**Strengths:**

The empirical evaluation is described in detail and takes care of some basic causes of evaluation bias.

**Weaknesses:**

1. I do not think the paper evaluates what it says, nor the paper is convincing enough in supporting the claims. The paper claims that 1 phase and 2 phase heuristic algorithms produced through the process involving LLM demonstrate better performance than the baselines. However, the baselines are undergraduate  textbook algorithms, there is plenty of literature on improving over these textbook algorithms, and the LLM used was likely trained on publications introducing those improvements. There are several ways to deal with this in evaluation:

* either include the most advanced SOTA non-machine-learned algorithms for 2d bin packing (there are works on this, and their citations are deliberately omitted from the submission) and show that LMM produces better solutions;
* or demonstrate that the solutions produced by LLMs are not variants of published solutions of 2d bin packing (which is harder to do);
* or provide the SOTA algorithms themselves as the input prompts and ask the LLM to improve them without changing the overall logic.

None of this is done in the paper. Without the above, I am inclined to think that the LLM used what it learned from the literature on 2d bin packing to recognized the baselines and return variants of advanced heuristic algorithms from the literature. I am not saying that it is what happened (because I do not have enough information), but the paper does not make any attempt to rule this out.

2. The paper uses the term 'optimal' loosely.  Lines 358-361 say that LLM achieved 'optimal' solutions within 2 iterations. An optimal solutation to 2d bin packing problem takes at least exponential time in the problem size unless P=NP. Either LLM returns a trivial optimal solution which is not practical at all, or the solution is not optimal. In either case, it is not clear at all what the claim in lines 358-361 states. I suspect that the intent is that LLM converged to a solution in two iterations. I am afraid this reinforces my concern in the previous point that it is more of pattern matching through the existing literature than constraint recognition and algorithmic capabilities of LLMs.

3. The datasets are said to be randomly generated. This is a highly structured problem, so 'randomly' does not necessarily means 'high entropy' or 'uniform'. Without knowing how the random generation was implemented, it is impossible to conclude that the LLM learned how to solve the 2d bin packing problem  (even if my concerns in points 1 and 2 are futile) rather  than learning the manifold of randomly generated instances, which is very likely to be rather thin and low-dimensional for this problem.

**Questions:**

1. Talking about optimality: the theoretical best possible approximation is 1.5 multiplicative for a polynomial time algorithm, if I remember correctly. It is possible to estimate approximation of LLM generated algorithms. How close are they to 1.5? What is the approximation of baselines? What is the best approximation of human-designed algorithms in the literature?

2. There are classes of instances of different approximation hardness for the 2d bin packing problem, regarding of the problem size. This is available in the literature. It is quite possible that your evaluation suffers from Simpson paradox. Could you please report approximation estimate of learned solutions for each approximation class separately?

---

### Official Review · Reviewer_6zvQ · 2025-11-02

**Soundness:** 2
**Presentation:** 2
**Contribution:** 2
**Rating:** 2
**Confidence:** 4

**Summary:**

This paper presents a systematic evaluation framework for assessing the capabilities of Large Language Models (LLMs) in solving the 2D bin packing problem, a combinatorial optimization problem. By integrating LLM-generated heuristics within an iterative, evolutionary approach, the authors demonstrate that models like GPT-4o can effectively generate and refine solutions, outperforming traditional heuristics such as Finite First-Fit and Hybrid First-Fit in terms of solution quality and efficiency. The framework enables comprehensive assessment of LLM reasoning, learning, and adaptation in optimization tasks, contributing valuable benchmarks and methodologies for future research in AI-driven combinatorial optimization.

**Strengths:**

- Interesting topic and clear writing

**Weaknesses:**

- The evaluation framework and results are primarily focused on the 2D bin-packing problem, a specific NP-hard problem. The insights and methods may not readily generalize to other combinatorial optimization problems or real-world industrial applications, limiting the broader impact of the work.

- The experiments are conducted on relatively small problem instances (e.g., 50 items, bins of fixed size). The approach’s scalability to larger, more complex, and industrial-scale problems is not thoroughly demonstrated, raising questions about its practical applicability.

**Questions:**

- How well do you expect your evaluation framework and the LLM-based heuristic generation approach to generalize to other combinatorial optimization problems beyond 2D bin packing? Have you considered or tested the applicability to problems such as vehicle routing, scheduling, or higher-dimensional packing?

- How does the performance of your LLM-generated heuristics compare against more recent or advanced ML-based approaches, such as deep reinforcement learning methods or hybrid algorithms optimized for larger or more complex instances?

---

### Note · Program_Chairs · 2026-01-17
**Submission Desk Rejected by Program Chairs**

The following references in this submission do not refer to real documents and/or have major errors in bibliographic information:

 José F Oliveira, Maria A Carravilla, and Fabio Furini. Two-dimensional cutting and packing problems: a survey. International Transactions in Operational Research, 2023.